# Multiferroic Coupling of Ferromagnetic and Ferroelectric Particles through Elastic Polymers

**DOI:** 10.3390/polym14010153

**Published:** 2021-12-31

**Authors:** Liudmila A. Makarova, Danil A. Isaev, Alexander S. Omelyanchik, Iuliia A. Alekhina, Matvey B. Isaenko, Valeria V. Rodionova, Yuriy L. Raikher, Nikolai S. Perov

**Affiliations:** 1Faculty of Physics, Lomonosov Moscow State University, 119991 Moscow, Russia; isaev.danil@gmail.com (D.A.I.); ya.alekhina@physics.msu.ru (I.A.A.); mathew.isaenko@gmail.com (M.B.I.); perov@magn.ru (N.S.P.); 2Institute of Physics, Mathematics & IT, Immanuel Kant Baltic Federal University, 236041 Kaliningrad, Russia; asomelyanchik@kantiana.ru (A.S.O.); VVRodionova@kantiana.ru (V.V.R.); 3Institute of Natural Sciences and Mathematics, Ural Federal University, 620000 Ekaterinburg, Russia; raikher@icmm.ru

**Keywords:** ferromagnetic particles, ferroelectric particles, magnetoelectric composite, multiferroics, elastic properties

## Abstract

Multiferroics are materials that electrically polarize when subjected to a magnetic field and magnetize under the action of an electric field. In composites, the multiferroic effect is achieved by mixing of ferromagnetic (FM) and ferroelectric (FE) particles. The FM particles are prone to magnetostriction (field-induced deformation), whereas the FE particles display piezoelectricity (electrically polarize under mechanical stress). In solid composites, where the FM and FE grains are in tight contact, the combination of these effects directly leads to multiferroic behavior. In the present work, we considered the FM/FE composites with soft polymer bases, where the particles of alternative kinds are remote from one another. In these systems, the multiferroic coupling is different and more complicated in comparison with the solid ones as it is essentially mediated by an electromagnetically neutral matrix. When either of the fields, magnetic or electric, acts on the ‘akin’ particles (FM or FE) it causes their displacement and by that perturbs the particle elastic environments. The induced mechanical stresses spread over the matrix and inevitably affect the particles of an alternative kind. Therefore, magnetization causes an electric response (due to the piezoeffect in FE) whereas electric polarization might entail a magnetic response (due to the magnetostriction effect in FM). A numerical model accounting for the multiferroic behavior of a polymer composite of the above-described type is proposed and confirmed experimentally on a polymer-based dispersion of iron and lead zirconate micron-size particles.

## 1. Introduction

Multiferroics are the substances with two or more ‘ferro’ properties (ferromagnetic, ferroelectric, ferroelastic), integrated in a single entity. Among other new smart materials, they are extensively studied due to their wide prospects in sensing and energy transforming and harvesting techniques [1,2,3] since such working elements are enticing due to their ability to operate in wide frequency, field, temperature, etc., ranges [1,2,3,4,5,6,7,8,9]. The point of highest interest in multiferroics is their ability to perform magnetoelectric conversion. Namely, either the direct magnetoelectric effect that is an electrical polarization under magnetic field $\Delta P=P{\left(E\right)}_{H}-P{\left(E\right)}_{0}$, or the inverse one that is a magnetization change under an electrical field $\Delta M=M{\left(H\right)}_{E}-M{\left(H\right)}_{0}$.

The magnetoelectric effect (MEE) in composite multiferroics exceeds this effect in natural multiferroic materials [4]. The highest values of MEE are obtained on solid composite multiferroics, e.g., layered materials or nanocomposites, which comprise both magnetostrictive and electrostrictive (piezoelectric) phases [5,6]. An instructive example is a composite that consists of a closely-packed and sintered mixture of ferromagnetic (FM) and ferroelectric (FE) grains. As the grains are in tight contact, restriction of one component of the composite caused by the respective external field transforms into mechanical stress that acts on the other component inducing a piezo effect.

In composite multiferroics of another sort, the FM or/and FE particles are bound by a piezopolymer [7]. The piezo effect in the matrix increases the interphase interaction and, due to that, enhances the MEE. Note, however, that this enhancement amounts to just a small part of the major MEE [8,9]. In fact, the main merit of polymer multiferroics in comparison with solid nanocomposites is their ability to deform without destruction that makes them available for flexible electronics. Moreover, by employing biocompatible polymer matrices, it is possible to expand applications of such systems to biomedicine [7].

The polymer multiferroic composites in soft matrices make a special case since in these systems the particle displacements might become quite substantial even under the fields of moderate strength. The closest to the polymer FM + FE composites under discussion are well-known magnetoactive elastomers (MAEs) [10,11,12,13,14,15,16,17,18,19,20]. In a soft MAE, an applied magnetic field, acting on the particles, induces the forces that are strong enough to considerably change the spatial distribution of particles and even produce macroscopic deformations of an MAE sample [21].

In Refs. [22,23], to produce a material capable of MEE, an MAE based on a silicone-rubber matrix and FM (NdFeB or BaFe_12_O_19_) micron-size particles was modified by adding to it (prior to polymerization) some amount of FE (lead zirconate, PbZr_x_Ti_1–x_O_3_) particles of micron-size dispersity as well. Note that due to the high compliance of the matrix, the field-induced motion of the FM particles in such a material is accompanied by the magnetodeformational effect [24,25].

Since the composites such as MAEs do not need to be detailed at the real molecular level, a model approach that is successfully applied to simulate such systems is the coarse-grained molecular dynamics [26,27]. In this framework, the interparticle magnetic dipole–dipole interaction is taken in the classical approximation assuming each dipole moment to be pointwise and located in the center of the particle.

In the present study, the molecular dynamics method is applied to a polymer multiferroic, viz. a non-piezoelectric soft matrix filled with FM and FE particles. The particles as well as the polymer beads are treated as elementary structureless entities (‘molecules’) connected by ‘virtual springs’. Evidently, such a model should display the MEE as it allows for the elastic interaction between the particles that are also coupled by magnetic and electric dipole–dipole interactions.

## 2. Materials and Methods

### 2.1. Numerical Model

The model composite is the polymeric matrix with embedded microparticles of two kinds: FM and FE ones. The matrix itself is assumed to be an assembly of electromagnetically neutral beads. All the elements—both particles and beads—are spherical objects. All of the polymer beads are identical with a radius of 2.5 μm. Both FM and FE particles are polydisperse with sizes distributed lognormally: mean radius R=5 μm, standard deviation σ=0.7. The size distribution cutoff at radius 5 μm was used to avoid appearance of extremely large particles. The respective particle and bead numbers together with their volume fractions and other material parameters are given in Table 1. The magnetization of FM particles is associated with that of bulk iron, and spontaneous polarization of FE particles is set equal to the corresponding characteristic of a typical bulk lead zirconate.

The simulated sample has the dimensions of 400 × 400 × 100 µm^3^, shaped as a rectangular cuboid, and all of its faces are fixed (immovable). The particles and beads are distributed randomly over the sample; a proximity check is used to prevent particle overlapping. The elements of the ensemble are connected by identical springs, thus forming an elastic network; each element is linked to all the others inside a surrounding sphere of reference radius 4R.

The FE particles bear permanent magnetic moments $\vec{m}$ of respective magnitudes $m=M_{s}V$, where $M_{s}$ is the saturation magnetization and V the particle volume. Magnetic moments are assumed to be located in the geometry centers of the particles and are free to rotate with respect to their bodies. In other words, each FM particle is a single-domain and magnetically isotropic. Therefore, under an applied field $\vec{H}$, the particle always magnetizes in its direction. The FE particles are ascribed similar properties and behavior: they bear permanent electric moments $\vec{e}$ of magnitudes $e=P_{s}V$, where $P_{s}$ is the saturation polarization, and are intrinsically isotropic. Accordingly, each FE particle always aligns its dipole moment with the applied field $\vec{E}$.

All of the links in the system are ‘virtual’ Hookean springs with common stiffness coefficient k and varying equilibrium lengths l; the distribution of l’s is evaluated after the initial placement of all the particles and beads inside the sample is accomplished. The overall Young’s modulus of the polymer matrix relates to the elastic parameter k and the mean value $\bar{l}$ as
(1)$$Y=k\bar{l}/S=k\bar{l}/\left(\pi R^{2}\right),$$
where $S=\pi R^{2}$ is the reference cross-section area of a particle. In the simulation, the value of Y is set to 10 kPa.

External DC uniform magnetic or/and electric fields (${\vec{H}}_{0}$ and/or ${\vec{E}}_{0}$) are applied to the sample along the *Oz* direction that is the shortest dimension of the sample; if in combination, the fields are coaligned. When performing simulations, the external fields are used in the form ($\vec{H}$ and/or $\vec{E}$), meaning that those are internal fields, i.e., renormalized with regard to the respective depolarizing effects; the nonuniformities induced by edge effects are neglected. Therefore, the field experienced by a particle magnetic moment includes two contributions: $\vec{H}$ induced by an externally applied field and another, ${\vec{H}}_{d}$, that is the joint contribution of all the magnetic neighbors. For an FE particle, this is a sum of $\vec{E}$ and ${\vec{E}}_{d}$. Whereas the former is assumed to be uniform, the latter is essentially non-uniform.

The force acting on a respective particle is a sum of two contributions. One is due to the effective local field whereas the other accounts for the elastic forces on the part of the elastic links with the neigbors:(2)$${\vec{F}}_{FM}={\vec{F}}_{m}+{\vec{F}}_{el}=\left(\vec{m}\nabla \right){\vec{H}}_{d}+\underset{i}{\sum }\pi Y\frac{R^{2}}{l_{i}}{\vec{\;\delta l}}_{i};$$
(3)$${\vec{F}}_{FE}={\vec{F}}_{e}+{\vec{F}}_{el}=\left(\vec{e}\nabla \right){\vec{E}}_{d}+\underset{i}{\sum }\pi Y\frac{R^{2}}{l_{i}}{\vec{\;\delta l}}_{i},$$
where ${\vec{\delta l}}_{i}$ is the change of length ${\vec{l}}_{i}$ of the ‘virtual spring’ between neighboring particles. Figure 1 illustrates the scheme of interaction of the particles of different kinds in the model composite.

The above-described model extends the one described earlier in [15]. The novelty of this model is that it considers 3D samples instead of 2D and incorporates the dipole–dipole interactions between FE particles into the scheme.

The treatment of the interparticle fields is done in the following way. As the dipole–dipole field decreases as a cube of the interparticle distance, we neglect the contributions from the particles located farther than 6R from the center of a given one; i.e., a particle interacts with 10 neighbors on average. On the other hand, due to the presence of polymer beads, the FM and FE particles cannot approach each other closer than ~4.5R.

The energy function of the model sample is (in SI system):(4)$$U=\underset{m}{\sum }\pi Y\frac{R^{2}}{l_{m}}{\vec{\delta l}}_{m}^{2}+\frac{1}{4\pi \mu_{0}}\underset{ij}{\sum }\left[\frac{\left({\vec{m}}_{i}{\vec{m}}_{j}\right)}{l_{ij}^{3}}-3\frac{\left({\vec{m}}_{i}{\vec{l}}_{ij}\right)\left({\vec{m}}_{j}{\vec{l}}_{ij}\right)}{l_{ij}^{5}}\right]+\frac{1}{4\pi \varepsilon_{0}}\underset{kl}{\sum }\left[\frac{\left({\vec{e}}_{k}{\vec{e}}_{l}\right)}{l_{kl}^{3}}-3\frac{\left({\vec{e}}_{k}{\vec{l}}_{kl}\right)\left({\vec{e}}_{l}{\vec{l}}_{kl}\right)}{l_{kl}^{5}}\right]-\underset{i}{\sum }({\vec{m}}_{i}{\vec{H}}_{i})-\underset{j}{\sum }({\vec{e}}_{j}{\vec{E}}_{j}),$$
which includes the elastic contribution, the dipole couplings of both kinds (for $\vec{m}$’s and $\vec{e}$’s) as well as the orientational interactions of the dipoles with the local fields of respective kinds; here, index m encompasses all elements of the ensemble, indices i and j refer to FM particles, whereas indices k and l to FE particles.

Coarse-grained molecular dynamics [28,29,30,31], the ‘virtual spring’ method [26,27], and the Verlet algorithm [32] were used to minimize function U, thus obtaining the equilibrium state of the sample under a given combination of electric and magnetic fields. By that, one is able to consider the system response under cyclic (but quasistatic) changes of both applied fields. Minimization was accomplished using the sliding window approach. For each iteration, the deviation of the current energy of the system from the average energy over the past 10 iterations was calculated. The convergence goal was 1% deviation.

The displacement of particles inside the sample volume was calculated using the Verlet algorithm. The following iterative formula was used for the changed position of each particle:(5)$${\vec{r}}_{i+1}=2{\vec{r}}_{i}-{\vec{r}}_{i-1}+\frac{1}{m}{\vec{F}}_{i}\Delta t^{2},$$
where *i* is the time step, ${\vec{r}}_{i}$ the particle position at the *i*-th step, ${\vec{F}}_{i}$ the sum of forces applied to the particle, m the particle mass, and Δt the time step, which was set to $10^{-8}$ s.

To analyze the emergence of MEE, the instantaneous state of the system was monitored by taking a series of snapshots of a box positioned inside the sample and fixed in space. The test box is a thin (along *Ox*) rectangular parallelepiped. As an example, in Figure 2, the superposition of projections of the particles inside the test box on the O*yOz* plane is shown. In this mode of presentation, the overlay of the particle projections along the line of vision does not at all imply that those particles are in close contact.

### 2.2. Experiment

The procedure of preparation of an elastomeric composite with FM and piezoelectric fillers was described earlier in [22]. In this work, the two-component silicone–rubber polymer was filled with carbonyl iron (GNIIChTEOS, Moscow, Russia) and lead zirconate titanate (PZT-19) particles (ELPA, Uglich, Russia). The mean size of both kinds of particles was ~5 μm. The particles were mixed with the first component of silicone compound and ultrasonicated. Then, the second component was added to the mixture, and the resulting mixture was homogenized by mechanical and ultrasonic steerings. The liquid composite was placed in the form with an antiadhesion surface and heated at 100 °C for 1 h. The total volume content of iron and PZT particles was 20%, and the volume ratio between the phases was 1:1. The Young’s modulus of the initial silicone was 10 kPa.

Magnetic measurements were carried out with VSM model 7400 by Lakeshore (Westerville, OH, USA) in the field range up to 1200 kA/m at room temperature (~290 K). To measure the hysteresis loop under external modulating electric field, the sample (rectangular parallelepiped with size 4 × 4 × 1 mm^3^) was placed between conductive plates that were connected to a high-voltage source [22]. The capacitor plates rigidly fixed the sample, so that its shape was virtually unchanged during the measurements. The value of external voltage was 5 kV, and the thickness of the sample was 1 mm between plates (E=5 MV/m). The applied electric and magnetic fields were collinear.

## 3. Results and Discussion

The model detailed in Section 2.1 was used to simulate the magnetization M(H) and electric polarization P(E) curves of the samples. In Figure 3a,b, these curves are shown for the situation when bias fields E and H were zero.

Electrical polarization curves P(E) under an applied (bias) magnetic field ranging from 16 to 400 kA/m were simulated; as an example, in Figure 3c, the results for H=400 kA/m are shown. The magnetization curves were simulated with the applied (bias) electric field varying between 5 and 90 MV/m; Figure 3d presents the obtained data for E=90 MV/m.

The differences between the biased and non-biased curves demonstrated in Figure 4 provide explicit evidence of the existence of direct and inverse MEEs in the composite. In other words, the occurring differences confirm the multiferroic nature of the studied composite. Indeed, the plots of Figure 3d show that if subjecting a sample to an electric field, one fixes the value of the latter and after that applies a co-aligned magnetic field, the polarization would change. Equally, if subjecting a sample to a magnetic field, one fixes the value of the latter and then applies a co-aligned electric field, the magnetization would change.

The undertaken simulation enables one to analyze the effects in more detail. In particular, Figure 3c shows that direct MEE may change its sign depending on the strength of the bias magnetic field. To clarify the issue, in Figure 4, the increments $\Delta P\left(E\right)=P{\left(E\right)}_{H}-P{\left(E\right)}_{0}$ and $\Delta M\left(H\right)=M{\left(H\right)}_{E}-M{\left(H\right)}_{0}$ are shown. As seen, negative increments are inherent for both types of MEE. Expectedly, the magnitude of the effects increased with the strength of the bias field. The indicated error ‘corridor’ was evaluated from the statistical standard deviation caused by the initial distribution of the FM and FE particles in the matrix.

Let us outline the qualitative situation. As mentioned, in a composite of the considered type—a mixture of iron and PZT micron-size particles in a magnetically/electrically neutral and mechanically soft (and thus compliant) matrix—the MEE is due to re-grouping of the particles. According to the assumption, the particles are both electrically and magnetically polarizable due to internal rotation of permanent dipole moments. Any of the externally applied fields aligns the particle dipole moments. This changes the dipole–dipole interaction and induces the forces—see Equations (2) and (3)—which strive to arrange the particles in the structures most favorably from the viewpoint of minimizing the collective energy of the assembly, the members of which they are. A soft matrix, although preventing the particles from unbound motion, nevertheless gives them a certain motional freedom. The change of spatial distribution means the reorganization of interparticle distances that, in turn, modify the dipole–dipole forces in the assembly.

The most important consequence that ensures a multiferroic property (a MEE) of the composite, is that the shifted particles (e.g., of FM type) generate inside the matrix a mesoscopically non-unform distribution of mechanical stresses. Due to its elasticity, the matrix transfers those to FE particles which, in turn, respond by electric polarization.

When a composite is subjected to a bias field of one type (electric/magnetic), the latter affects the initial particle distribution and prepares some pre-structured state which then begins to change under the action of the field of another type (magnetic/electric). The arrangement that the latter—below we term it the *acting* field—strives to ultimately organize, differs from that of the bias-induced pre-structure. Due to that, during the stage where the acting field is equivalently lower than the bias one, its effect is mostly spent on destroying the pre-structure; see negative cusps in Figure 4.

It is reasonable to infer that the field of a given kind when acting on a pre-structured system first and foremost works on the akin particles: $\vec{E}$ on FE and $\vec{H}$ on FM ones. In that way, the particles of the alternative kind and the springs attached to them work as obstacles. In the first stage, the acting field is quite weak and those obstacles make the polarization/magnetization lower than that of an unbiased system. The stronger the bias, the stronger the acting field required to gain full control of the system. Because of that, in Figure 4, the points where the increment dependencies cross the zero level move to stronger fields as the bias increases.

The positive cusps on the curves of Figure 5 mean that the system polarizes/magnetizes better than the same system in the absence of any bias. It seems that, upon having acquired well-favorable positions with respect to a sufficiently strong acting field (e.g., FM particles under a magnetic field), the particles form a set of elongated aggregates, i.e., a uniaxially ordered structure. This newly established arrangement facilitates the particles of the alternative phase to form their aggregates of the same type that is tempted by the bias field. Finally, under a very strong acting field, the magnetization/polarization saturates, and the difference between biased and unbiased systems becomes negligible. The changes in magnetization and polarization at maximum bias fields in the measuring field that range up to 2400 kA/m (ΔM) and 900 MV/m ΔP are presented in Appendix A).

As the polarization and magnetization of the particles of both kinds are limited by their respective saturation levels, the bias effect saturates as well. This is illustrated by Figure 5, where the values of extrema of the curves presented in Figure 4 are shown. As is seen, the bias fields H higher than 100 kA/m and E higher than 30 MV/m are not available for further enhancement of the MEEs. Notably, the saturation effect manifests itself with respect to both the negative and positive cusps of the curves such as those shown in Figure 5.

For comparison of the simulation and experiment, a set of measurements of the inverse MEE $\Delta M\left(H\right)=M{\left(H\right)}_{E}-M{\left(H\right)}_{0}$ was undertaken. Figure 6 shows the results obtained foron the sample biased by an electric field with athe strength of 5 MV/m. Besides the agreement by the order of magnitude, the non-monotonic character of the simulated and measured effects is evident.

## 4. Conclusions

Multiferroics based on a soft polymer and a mixture of ferromagnetic and ferroelectric microparticles have been studied. The fundamental mechanism of magnetoelectric coupling without direct strictive contact between FM and FE particles and without a piezopolymer in the composite matrix is suggested. This mechanism works via mutual displacement of particles of both types under external magnetic or electric fields: the alignment of particles of one type results in the pre-structured distribution of the particles of the alternative kind. This texturing changes the dipolar coupling between particles and thus affects their polarization. A structure model of a bead-spring type is constructed and used for simulations. The obtained numerical results are in qualitative agreement with the experiment performed on a sample of elastomers filled with iron and PZT particles. The developed model provides a better understanding of the physical mechanism of magnetoelectric coupling. The presented results can be used for the optimization of structural properties of composite materials to attain higher efficiency in magnetoelectric transformation aimed at various fields. In particular, polymer multiferroics have potential for biomedical applications, where the mutual influence of particle displacements under applied magnetic and/or electric fields accompanied by mechanical deformations of the matrix is required.

## Figures and Tables

**Figure 1 polymers-14-00153-f001:**
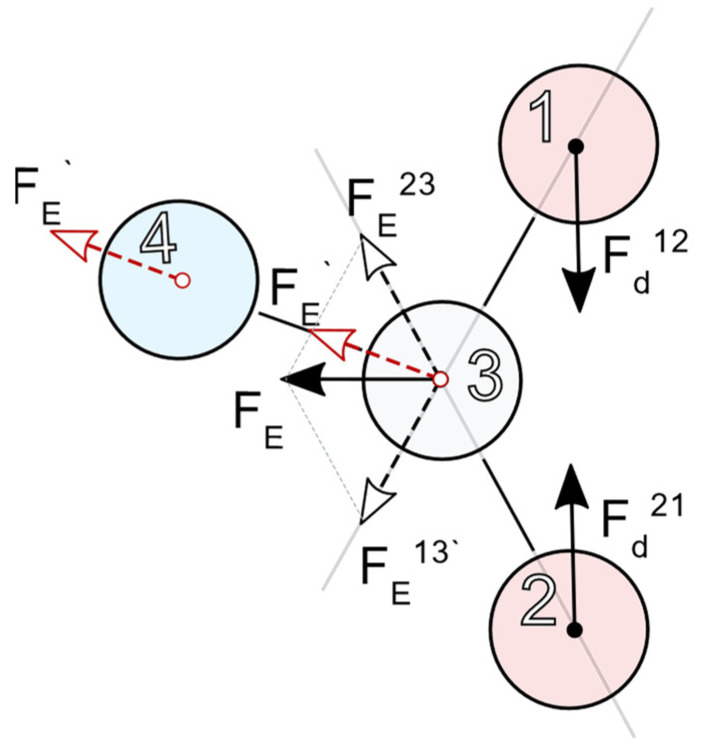
Diagram of the coupling hierarchy in a three-component composite. Particles 1 and 2 interact as magnetic or electrical dipoles and exert forces $F_{d}^{12}$ and $F_{d}^{21}$ on each other. This interaction causes elastic stresses which are transferred by spring links (solid lines) to the polymer particle 3 that, together with the springs attached to it, resembles the matrix. Due to the spring strains, particle 3 experiences elastic forces $F_{E}^{13}$ and $F_{E}^{23}$, the vector sum of which is transferred as a single force $F_{E}$ via the appropriate spring to particle 4 (the force projection is $F_{E}^{\text{′}}$ ) that transfers it outside the presented fragment of the system. We point out that $F_{E}^{\prime }$ is drawn twice to show its translation from particle 3 to particle 4.

**Figure 2 polymers-14-00153-f002:**
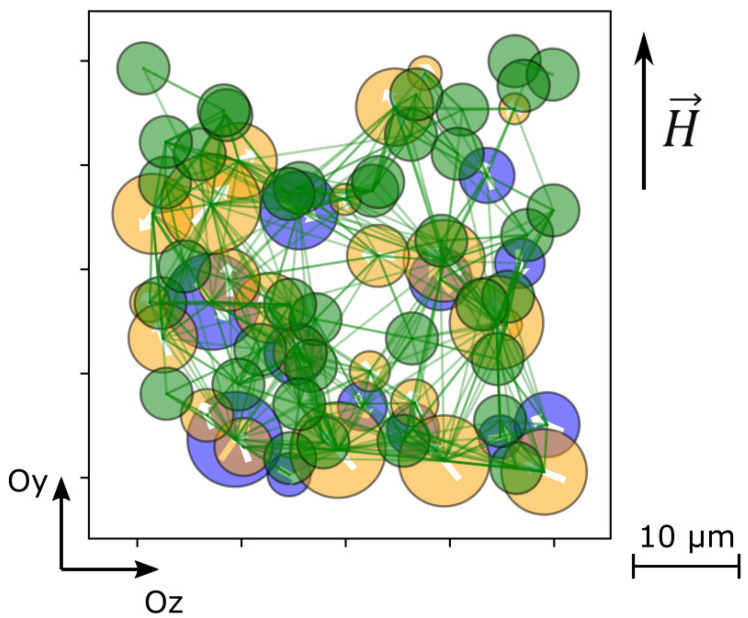
Snapshot of the test box; blue spheres are FM particles, orange ones are FE particles, white arrows show their magnetic/electric polarizations, green spheres are polymeric beads, green lines denote elastic links; the snapshot region is 10 × 50 × 50 μm along *Ox*, *Oy,* and *Oz* axes, respectively.

**Figure 3 polymers-14-00153-f003:**
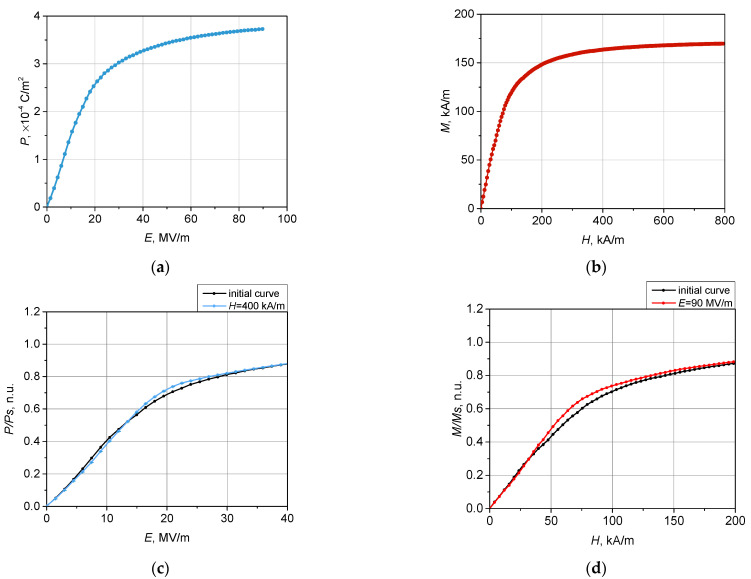
Simulated polarization (**a**) and magnetization (**b**) curves for a soft elastic matrix filled with a mixture of FM and FE particles. (**c**) Polarization curves under an external (bias) magnetic field: *H* = 0 and 400 kA/m; (**d**) magnetization curves under an external (bias) electric field: *E* = 0 and 90 MV/m.

**Figure 4 polymers-14-00153-f004:**
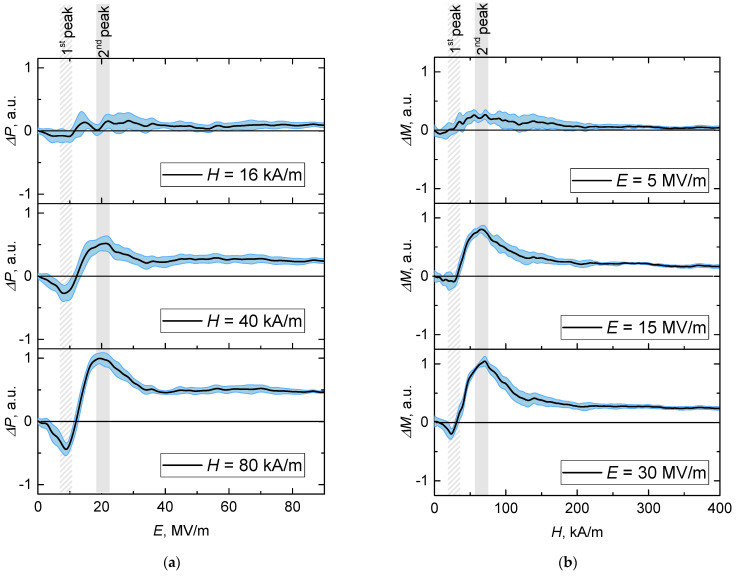
Simulated (**a**) direct $\Delta P\left(E\right)=P{\left(E\right)}_{H}-P{\left(E\right)}_{0}$ and (**b**) inverse $\Delta M\left(H\right)=M{\left(H\right)}_{E}-M{\left(H\right)}_{0}$ magnetoelectric effects; the vertical axis is scaled in arbitrary units.

**Figure 5 polymers-14-00153-f005:**
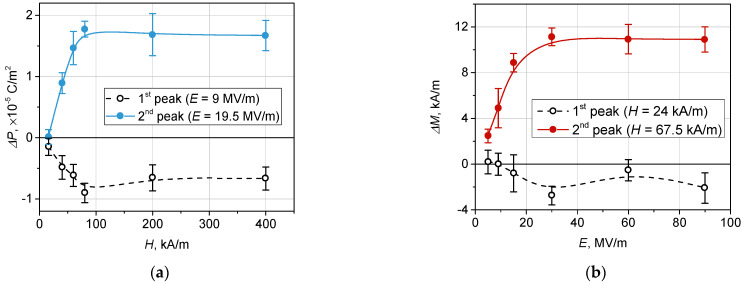
(**a**) Dependence of polarization increment ΔP on the strength of bias magnetic field strength; (**b**) dependence of magnetization increment ΔM on the strength of bias electric field.

**Figure 6 polymers-14-00153-f006:**
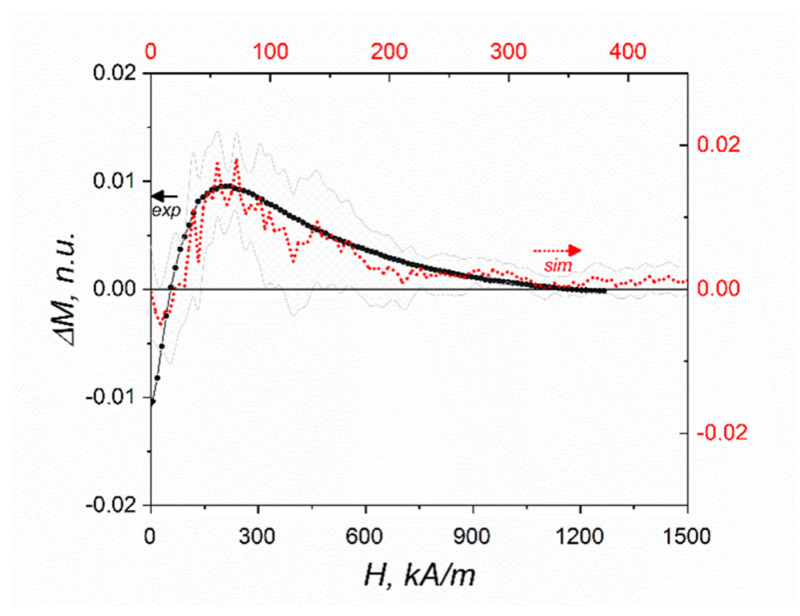
Magnetization increment $\Delta M\left(H\right)=M{\left(H\right)}_{E}-M{\left(H\right)}_{0},$ normalized units under an electric bias of 5 MV/m; experimental data (black) and simulation data (red) with error bars (grey).

**Table 1 polymers-14-00153-t001:** System parameters.

Type of Particles	Density, kg/m^3^	Volume Concentration, %	Number of Particles in the System	$\mathrm{Saturation}\;\mathrm{Magnetization}\;M_{s}$ $/\mathrm{Polarization}\;P_{s}$
Polymeric	1003	7	17113	–
Ferromagnetic	7874	10	9658	1700 kA/m
Ferroelectric	4700	10	9717	4000 mC/m^2^

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
