# Peer review of "Multiferroic Coupling of Ferromagnetic and Ferroelectric Particles through Elastic Polymers"

_polymers, 2021, doi:10.3390/polym14010153_

Round 1

Reviewer 1 Report

Authors are requested to revise section 2.2 (Experiment) part in more details so that one can reproduce the materials.

English Language is almost good but authors are requested to revise incoherence in sentences and simplify very long/complex sentences. 

Author Response

Rev. 1. Comment 1:

Authors are requested to revise section 2.2 (Experiment) part in more details so that one can reproduce the materials.

Authors’ response:

It is a justified request. Indeed, the description of experiment might seem quite brief. However, this is not an offhandedness, such a proportion was due to two circumstances. First, the focus of the work is on numerical modelling, and a reference to experiment is in a way auxiliary since the agreement is yet only qualitative. Second, an extended description of the experiment would not bring in any relevant news: the pertinent details are already published in our Ref. [20] which we point out right in this context.

That is why when revising Section 2.2, we still tried to keep it brief. But it is true that some info should be added. We have done that by extending Sec. 2.2 by the following paragraph.

“The mean size of the particles of both kinds was ~5 μm. The powders were mixed with the first component of the silicone compound and ultrasonicated. Then the second component was added, and the resulting mixture was homogenized by mechanical and ultrasonic steerings. The liquid composite was placed in a mould with antiadhesion surface and heated at 100 °C for 1 h.”

Rev. 1. Comment 2:

English Language is almost good but authors are requested to revise incoherence in sentences and simplify very long/complex sentences.

Authors’ response:

The text has been re-read carefully and re-checked with regard to the language. A number of errors and bugs removed.

Reviewer 2 Report

This work by L.A. Makarova et al. has reported the theoretical and experimental studies on multiferroic polymer composites composed of ferromagnetic and ferroelectric particles.  The findings are scientifically important for understanding the basics of magnetoelectric couplings. The manuscript is well-presented. I only have one concern: The Abstract or Conclusion sections should be revised to highlight the specific and important scientific findings. Except that, I would like to recommend this manuscript be published in polymers.

Author Response

We appreciate very much the opinion that:

This work by L.A. Makarova et al. has reported the theoretical and experimental studies on multiferroic polymer composites composed of ferromagnetic and ferroelectric particles. The findings are scientifically important for understanding the basics of magnetoelectric couplings. The manuscript is well-presented.

Rev. 2. Comment 1:

I only have one concern: The Abstract or Conclusion sections should be revised to highlight the specific and important scientific findings. Except that, I would like to recommend this manuscript be published in Polymers.

Authors’ response:

We have revised both the Abstract and Conclusions in order to emphasize the scientific findings of the work in the new version of the text.

Reviewer 3 Report

Dear Editor,

The manuscript “Multiferroic coupling of ferromagnetic and ferroelectric particles through elastic polymer” by Liudmila A. Makarova, Danil A. Isaev, Alexander S. Omelyanchik, Iuliia A. Alekhina, Matvey B. Isaenko Valeria V. Rodionova, Yuriy L. Raikher, Nikolai S. Perov studies numerically and experimentally a multiferroic material consisting of ferroelectric micrograins and ferromagnetic micrograins mixed with soft polymer material. Multiferroic and magnetoelectric  materials attract a lot of attention at the moment due to their unique properties and multiple applications that requires this kind of materials. In the current work the authors consider a magnetoelectric material in which magnetic field affects the electrical susceptibility and vice versa – the electric field affects magnetic susceptibility. Authors demonstrate this effect both in simulations and experiments. While the results looks interesting, their presentation is quite poor. Most of the work is devoted to numerical simulations. But the model is practically not introduced in the paper. It is very hard to understand what authors consider and therefore to understand if the results are correct. My suggestion is that authors first provide clear description of the model they consider and then the manuscript can be jugged.

  1. In Eq. (1) and (2) the vector dl is unclear. Each particle interacts with many other particles. Why the spring force is defined by a single vector? I expect there should be a sum over all neighboring particles?
  2. Is the particle 3 in Fig. 1 a polymer particle?
  3. The material model should be explained in more detail. It seems that there are three types of particles – polymer, FM and FE. What is the concentration of all of them? Is the spring constant the same for the springs connecting FE-FM and FE-FE and FM-Polymer and etc.?
  4. The system Hamiltonian should be shown. That would make understanding much easier.
  5. Why the torque acting on magnetic and FE particles is not taken into account?
  6. The mechanism of influence of magnetic field and electric field on particles is not clear. For example, the external fields do not enter into the force equations 1 and 2, since only inhomogeneous field create force.
  7. Single particle model is not provided. Does single magnetic particle have magnetic anisotropy? Are the particles perfect spheres (in simulations and experiment)? Does FE have anisotropy? What is the internal state of the FE particle? Is this a multidomain state? Of these are paraelectric particles?
  8. What is the sample shape in simulations and experiment?
  9. It seems that the magnetic field magnetizes the sample and it should change the shape. Probably, it should become elongated along the field direction to reduce the magnetic shape anisotropy. Similarly, the electric field should elongate the sample along the field axis. Are there dependence of the ME effect on the mutual orientation of external fields?
  10. That would be great if authors provide present phenomenological expression for the energy term describing this magneto-electric phenomenon.

Style:

  1. 4. Either caption or figures order is incorrect.
  2. In Fig. 1, the force FE is shown twice along the line connecting particles 3 and 4 and along the vector sum of forces Fe23 and Fe13. Which one is the right one?

Author Response

Rev. 3. Comment 1:

While the results look interesting, their presentation is quite poor. Most of the work is devoted to numerical simulations. But the model is practically not introduced in the paper. It is very hard to understand what authors consider and therefore to understand if the results are correct. My suggestion is that authors first provide clear description of the model they consider and then the manuscript can be judged.

Authors’ response:

We do agree that in the initial version the description of the numerical model was too short. In the new version, Section 2.1 (Numerical model) it is radically revised and extended with concern to give all the relevant details.

Detailed information with symbols and formulas is available in the attahced file.

Rev. 3. Comment 2:

In Eq. (1) and (2) the vector dl is unclear. Each particle interacts with many other particles. Why the spring force is defined by a single vector? I expect there should be a sum over all neighboring particles?

Authors’ response:

A just reproach. It is our fault that Eqs. (2) and (3) were not quite clear. Indeed, the force is a vectorial sum over all the forces imposed by the neighboring particles. Now the said equations are rewritten as (see attached file). i.e., the sums are presented explicitly.

Rev. 3. Comment 3:

Is the particle 3 in Fig. 1 a polymer particle?

Authors’ response:

Yes, particle 3 is a polymer bead. Now, this is mentioned in the figure caption.

Rev. 3. Comment 4:

The material model should be explained in more detail. It seems that there are three types of particles – polymer, FM and FE. What is the concentration of all of them? Is the spring constant the same for the springs connecting FE-FM and FE-FE and FM-Polymer and etc.?

Authors’ response:

The reviewer is right: there are three types of particles in the model. Their volume concentrations and other material parameters of the system are given in Table 1. In the adopted model, the properties of springs are the same for all types of particles. We corrected the description of the model in section 2.1 to clarify these details.

Rev. 3. Comment 5:

The system Hamiltonian should be shown. That would make understanding much easier.

Authors’ response:

No doubt, Reviewer 3 is true. Now the energy function is presented explicitly, see Eq.(4) and the following text in the new version.

Rev. 3. Comment 6:

Why the torque acting on magnetic and FE particles is not taken into account?

Authors’ response:

In the framework of this model, the particles (FE and FM) are spherical and intrinsically isotropic as it is now explained in Section 2.1 of the revised version. Due to those properties, the particles always polarize in the direction of the local field. When the field, changes the polarization readily follows it. Thus, no torques emerge in such an ensemble.

Rev. 3. Comment 7:

The mechanism of influence of magnetic field and electric field on particles is not clear. For example, the external fields do not enter into the force equations 1 and 2, since only inhomogeneous field creates force.

Authors’ response:

To a good extent, this response goes in line with that to Comment 6 in above. Since the particles are intrinsically isotropic, the magnetic and electrical polarization changes require zero energy (we neglect intraparticle dissipation). As the external field ( or ) is assumed to transform inside the sample in equally uniform acting field ( or ) they do not produce ant force acting on the particles. Therefore, in the problem under study, all the forces accounted for come from the dipole-dipole interactions: magnetostatic (between FM particles) and electrostatic (between FE particles). Now it is explicitly reflected in Eqs. (2) and (3) by introducing dipole fields  and .

Rev. 3. Comment 8:

Single particle model is not provided. Does single magnetic particle have magnetic anisotropy? Are the particles perfect spheres (in simulations and experiment)? Does FE have anisotropy? What is the internal state of the FE particle? Is this a multidomain state? Of these are paraelectric particles?

Authors’ response:

In the framework of this model, all particles are spherical as stated in the third line of Section 2.1. The single particle model is now described in much more detail in that Section, see the text after Table 1. The essence is that a particle of either type (FM or FE) is assumed to be internally isotropic. This means that its dipole moment is free to rotate (no energy barriers) with respect to the particle body. On the other hand, the particle is considered as effectively single-domain since its dipole moment has a permanent magnitude, e.g., for a FM particle it is , where  is the saturation magnetization and  the particle volume.  In this aspect they might be conditionally viewed at as superparamagnetic or superparaelectric, respectively, with the condition that no thermal fluctuation are implied in those definitions.

Rev. 3. Comment 9:

What is the sample shape in simulations and experiment?

Authors’ response:

The sample size in the model was set to 400×400×100 µm, as is indicated now at the beginning of the second paragraph of Section 2.1. The size of the sample in the experiment was 4×4×1 mm. The latter note is added in Section 2. 2.

Rev. 3. Comment 10:

It seems that the magnetic field magnetizes the sample and it should change the shape. Probably, it should become elongated along the field direction to reduce the magnetic shape anisotropy. Similarly, the electric field should elongate the sample along the field axis. Are there dependence of the ME effect on the mutual orientation of external fields?

Authors’ response:

The simulated sample is a rectangular cuboid with fixed rectangular faces. This confinement is imposed to comply with the experiment where the sample was rigidly fixed between the capacitor plates, so that during the measurements its shape is virtually unchanged. That is why here we neglect any deformations of the sample under external fields.

Certainly, the Reviewer’s 3 remark is fully justified for a general case, there the sample shape and field-direction effects would obligatory turn up. In further work, these effects would be taken into account, such a problem is among the first in our “to do” list.

As in the present work the directions of the fields are fixed, here the shape effect just means that the field strengths are those of acting internal but not external fields.

Rev. 3. Comment 11:

That would be great if authors provide present phenomenological expression for the energy term describing this magneto-electric phenomenon.

Authors’ response:

Here we work out a mesoscopic description of the magneto-electric effect. As such, our approach is alternative to the conventional phenomenology (like that in Landau & Lifshitz Electrodynamics of Continuous Media or Maugin Continuum Mechanics of Electromagnetic Media) where all the specifics of a polymer multiferroic is accounted for with the aid of a few macroscopic coefficients. In general view, the mesoscopy, as it goes in much more details of the material structure, provides a benchmark for the validity of phenomenological models, like expansions of the energy in powers of the macroscopic thermodynamic variables: magnetization, electric polarization, strain, etc.

Rev. 3. Comment 12:

Style: Either caption or figures order is incorrect.

Authors’ response:

We thank Reviewer 3 for pointing out this flaw. Now it is corrected.

Rev. 3. Comment 13:

Style: In Fig. 1, the force FE is shown twice along the line connecting particles 3 and 4 and along the vector sum of forces Fe23 and Fe13. Which one is the right one?

Authors’ response:

Both red dashed vectors are valid. The translation of  from the center of particle 3 to that of particle 4 emphasizes the idea of the model. This is explained in the re-written legend to Fig. 1, see the new version. Now the last part of the legend reads: “Due to the spring strains, particle 3 experiences elastic forces  and , the vector sum of which is applied as a single force  to the appropriate spring 3-4. This spring transfers  as projection  to particle 4 and via it outside of the presented fragment of the system; to point that out,  is drawn twice: at its source and at its destination.

Reviewer 4 Report

Dear authors

I have overall enjoyed article reading. The topic discussed by the authors is interesting to the audience. But several changes must be addressed before article acceptance.

It called my attention that the experimental setup was the same as previously reported in Ref. 20. What are the main differences between Ref. 20 and the results reported in this manuscript?

Major changes

Please rewrite the abstract. Describe the motivation for article writing, and present an overview of the experimental results and simulation. The main findings must be also described.

Lines 77 through 81: This paragraph describes the experimental setup for the simulation. Please include a sketch that summarizes particle allocation and spatial distribution. Is Figure 1 the corresponding sketch for particle allocation? if so, then mention Figure 1 in this paragraph.

Line 91: What about Van der Waal’s forces? Are they worth-considering in the simulation?

Line 114: What are the authors ‘criteria for using this distribution over another? Was this setup used somewhere else in specialized literature? Please provide more details

Line 124: Do particle collisions are expected in real materials? Have you performed SEM analysis of the assembled material? This may help you determine if particle collision are worth considering in the simulation. Please discuss about the validity of the simulation to approach real results.

Line 156: Papers should be self-contained, therefore, the details of the experimental setup must be here provided.  

Minor changes

Line 32: Please provide a brief description of the magnetoelectric effect. This journal is mainly focused on polymers, and magnetic effects must not be fully recognized by readers.

Line 64 through 67: This sentence is just too long, please considering splitting.

Line 80: within is misspelled.

Line 88: It is a good practice that before enumerating items, a sentence is included followed by “:”, e.g. in this work the following considerations were done: (i) the scheme…

Line 180: A pronoun is missing before the verb “establish”.

Author Response

It is utterly flattering to read that:

I have overall enjoyed article reading. The topic discussed by the authors is interesting to the audience.

Rev. 4. Comment 1:

It called my attention that the experimental setup was the same as previously reported in Ref. 20. What are the main differences between Ref. 20 and the results reported in this manuscript?

Authors’ response:

The point is that this paper is focused on a theoretical model, the experimental data is indeed not new, it is employed here just to show that the developed model is consistent with the measurements. Therefore, by no means the already published experiment is an issue of priority or novelty in this paper, its function here is auxiliary. That is why we do not at all go into experimental details referring to our Ref. [20] instead.

Rev. 4. Comment 2:

Major changes. Please rewrite the abstract. Describe the motivation for article writing, and present an overview of the experimental results and simulation. The main findings must be also described.

Authors’ response:

We agree with Reviewer 4. In compliance with his/her remark (that coincides with Comment 1 of Reviewer 2), we have extended Abstract and Conclusion parts.

Rev. 4. Comment 3:

Lines 77 through 81: This paragraph describes the experimental setup for the simulation. Please include a sketch that summarizes particle allocation and spatial distribution. Is Figure 1 the corresponding sketch for particle allocation? If so, then mention Figure 1 in this paragraph.

Authors’ response:

We clarify this issue in the text of the new version: lines 96-98. However, a reference to Fig. 1 does not seem appropriate here. For one thing, Fig. 1  sketches a fragment of the model assembly where polymer is treated as a set of beads that is, of course, not the case for a real sample.

Rev. 4. Comment 3:

Line 91: What about Van der Waal’s forces? Are they worth-considering in the simulation?

Authors’ response:

The van der Waals forces are very important here but implicitly. In real samples they are responsible for keeping the particles fully adhered to the polymer matrix. In the considered model full adhesion is equivalent to a statement that no one of the springs may disconnect form its site. It means that the van der Waals forces in the system are much stronger than any elastic ones experienced by the model springs.

Rev. 4. Comment 4:

Line 114: What are the authors ‘criteria for using this distribution over another? Was this setup used somewhere else in specialized literature? Please provide more details

Authors’ response:

The lognormal distribution of particle size is the most common in the literature (and the most successful one) to approximate the particle histogram in real systems (see, for example, Söderlund et al. ‘Lognormal size distributions in particle growth processes without coagulation’ Phys. Rev. Lett. 80 (1998) 2386., 10.1103/PhysRevLett.80.2386).

Rev. 4. Comment 5:

Line 124: Do particle collisions are expected in real materials? Have you performed SEM analysis of the assembled material? This may help you determine if particle collision are worth considering in the simulation. Please discuss about the validity of the simulation to approach real results.

Authors’ response:

We did not take into account possible particle collisions (does Reviewer 4 means overlapping or sticking?) both in experiment (after solidification of the sample) and simulation. The FM particles used practically did not stick together in the liquid carrier matrix. Some aggregates can exist, but after sample preparation no additive aggregates can arise. Therefore, these particles are distributed by and large uniformly in the polymer material. In the model, particle collisions (we mean coming in tight contact) were prevented by the presence of sufficiently stiff springs. Unfortunately, our lab facilities do not provide an opportunity perform a SEM study in short time.

Rev. 4. Comment 6:

Line 156: Papers should be self-contained, therefore, the details of the experimental setup must be here provided.  

Authors’ response:

Section 2.2. was extended and now it contains all details of experimental setup. Namely:

“The procedure of preparation of an elastomeric composite with FM and piezoelectric fillers was described earlier in [20]. In this work, the two-component silicone-rubber polymer was filled with carbonyl iron and lead zirconate titanate (PZT-19) particles (ELPA). The mean size of both kinds of particles was ~5 μm. The particles were mixed with the first component of silicone compound and ultrasonicated. Then the second component was added to the mixture and the resulting mixture was homogenized by mechanical and ultrasonic steerings. The liquid composite was placed in the form with antiadhesion surface and heated at 100 °C for 1 h. The total volume content of iron and PZT particles was 20%, the volume ratio between the phases was 1:1. The Young modulus of the initial silicone was 10 kPa.”

Rev. 4. Comments 7– 11. Minor changes

Line 32: Please provide a brief description of the magnetoelectric effect. This journal is mainly focused on polymers, and magnetic effects must not be fully recognized by readers.

Authors’ response:

In the revised version the very first paragraph of Introduction is devoted to explanation of the essence of magnetoelectric effects, both direct and inverse. (Lines 40-42)

Line 64 through 67: This sentence is just too long, please considering splitting.

Authors’ response:

Thanks for noting that. Done.

Line 80: within is misspelled.

Authors’ response:

Thanks again. Corrected.

Line 88: It is a good practice that before enumerating items, a sentence is included followed by “:”, e.g. in this work the following considerations were done: (i) the scheme…

Authors’ response:

We thank the reviewer for this comment. We rephrased the mentioned sentence as:

“The above-described model extends the one described earlier in [14]. The novelty of this model is that it considers 3D samples instead of 2D and incorporates the dipole-dipole interactions between FE particles into the scheme.”

Line 180: A pronoun is missing before the verb “establish”.

Authors response:

We are much obliged and apologize for the bug. The sentence is corrected now.

Round 2

Reviewer 4 Report

Mandatory changes have been addressed one-by-one. Therefore, the article can be published in present form